# Multitask Learning Strategy with Pseudo-Labeling: Face Recognition, Facial Landmark Detection, and Head Pose Estimation

**DOI:** 10.3390/s24103212

**Published:** 2024-05-18

**Authors:** Yongju Lee, Sungjun Jang, Han Byeol Bae, Taejae Jeon, Sangyoun Lee

**Affiliations:** 1School of Electrical and Electronic Engineering, Yonsei University, 50 Yonsei-ro, Seodaemun-gu, Seoul 03722, Republic of Korea; paulyongju@yonsei.ac.kr (Y.L.); jeu2250@yonsei.ac.kr (S.J.); 2School of Computer Science and Engineering, Kunsan National University, 558 Daehak-ro, Gunsan 54150, Jeollabuk-do, Republic of Korea; hbbae@kunsan.ac.kr; 3Taejae JeonMX Division, Samsung Electronics Co., Ltd., Suwon 16677, Gyeonggi-do, Republic of Korea; jtj7587@gmail.com

**Keywords:** multitask learning, face recognition, facial landmark detection, head pose estimation, pseudo-labeling

## Abstract

Most facial analysis methods perform well in standardized testing but not in real-world testing. The main reason is that training models cannot easily learn various human features and background noise, especially for facial landmark detection and head pose estimation tasks with limited and noisy training datasets. To alleviate the gap between standardized and real-world testing, we propose a pseudo-labeling technique using a face recognition dataset consisting of various people and background noise. The use of our pseudo-labeled training dataset can help to overcome the lack of diversity among the people in the dataset. Our integrated framework is constructed using complementary multitask learning methods to extract robust features for each task. Furthermore, introducing pseudo-labeling and multitask learning improves the face recognition performance by enabling the learning of pose-invariant features. Our method achieves state-of-the-art (SOTA) or near-SOTA performance on the AFLW2000-3D and BIWI datasets for facial landmark detection and head pose estimation, with competitive face verification performance on the IJB-C test dataset for face recognition. We demonstrate this through a novel testing methodology that categorizes cases as soft, medium, and hard based on the pose values of IJB-C. The proposed method achieves stable performance even when the dataset lacks diverse face identifications.

## 1. Introduction

Face analysis has improved considerably in various tasks, including detection [1,2], recognition [3,4,5], and estimation [6,7], with several models achieving remarkable accuracy and efficiency. This progress is attributed to the development of various network structures and the availability of a large number of face datasets for training. Although sufficient datasets are available for major tasks such as face recognition and face detection, substantial gaps exist in data availability for tasks requiring sensitive labeling, such as facial landmark detection and head pose estimation. This difficulty is especially noticeable in datasets with few face identifications (IDs) or a limited total amount of data, making it challenging to create robust and generalized models for the above-mentioned tasks.

Various methods have been explored to address the issue of the scarcity of face IDs within datasets. Typical single-task learning methods have been developed to overcome this challenge by finding commonalities between datasets or using information from other tasks. In tasks such as heterogeneous face recognition [8,9,10], style embedding features are learned to extract only common features from datasets in different domains with the same face class, or real faces are generated in different domains via generative models and then face recognition is performed to determine the identity. In studies of facial expression or stress recognition [11,12], small changes in facial landmarks have been used to specify the region to focus on, resulting in a more accurate estimate. Efforts towards domain adaptation have been made to address the challenge of data imbalance, which has been a significant problem regarding the effective functioning of facial landmark detection and head pose estimation in real-world scenarios [13,14,15,16,17,18]. Multitask learning has also been used to solve the problem of the lack of face IDs. A multitask cascaded convolutional neural network (MTCNN) [19] combines face detection and facial landmark detection to provide more accurate facial detection and alignment. To improve the facial recognition performance, some studies [20,21] have implemented multitask learning using facial attributes such as gender, hair color, and facial expressions.

However, these approaches assume label availability for multiple tasks within a dataset. Consequently, they inherently suffer from challenges as a fundamental solution to the lack of face IDs and overall volume in datasets. Alternative approaches exist, such as improving the data quantity through research in pseudo-labeling methods [22,23,24,25,26]. Inspired by multitask learning methods such as the all-in-one CNN [27] and SwinFace [21], as well as pseudo-labeling methods [22,23], such as those proposed by Pan et al., we hypothesize that the use of datasets that have numerous images with a variety of face IDs would be feasible for a task with a small number of face IDs. When combining datasets with different numbers of face IDs, we posit that the performance of precise pseudo-labeling can enable robust and generalized usability for any integration task in real-world scenarios. Figure 1 shows the overall pseudo-labeling and multitask learning framework. We present pseudo-labeling and multitask learning methods for face recognition, head pose estimation, and facial landmark detection to validate our assumptions.

To achieve precise pseudo-labeling, we prepared three pretrained networks: one pretrained multitask network for facial landmark detection and head pose estimation trained on the 300W-LP dataset and two pretrained networks for face recognition trained on the VGGFace2 [28] and MS1MV3 [29] datasets, respectively. First, we designed our pseudo-labeling framework by referring to 300W-LP [22], a previous labeling framework for facial landmarks and head pose estimation. In 300W-LP [22], the authors used iterative fitting to find the optimal labels because, when relabeling the 300W [30] dataset to 300W-LP, it is not possible to label manually. In contrast, in this study, instead of using iterative fitting, we designed a sampling and fitting algorithm for various views of the image and found the optimal label. In our pseudo-labeling framework, to reduce the bias of the existing pretrained networks and for more robust feature learning, the landmark pseudo-label was supplemented using an alignment method of eye landmarks based on eye blink detection. Furthermore, given that most facial analysis datasets contain images of celebrities, we used two pretrained networks for redundant validation through pseudo-facial identification. Finally, we produced a pseudo-labeled training dataset (PLTD) for multitask learning.

In the following sections, we describe how to perform multitask learning using the pseudo-labeled training dataset and propose two new loss functions for learning that complement each other without converging to a single task. By using our proposed pseudo-labeling and multitask learning frameworks, we show that the robustness and generalization of our model is increased as the number of face IDs and background noise in the dataset increases, and we can achieve similar or better performance than other state-of-the-art (SOTA) methods in each task. Our contributions can be summarized as follows.

We applied our pseudo-labeling framework to tasks lacking a face ID, such as facial landmark detection and head pose estimation. We also used an inherently large face ID face recognition dataset.Within our pseudo-labeling framework, we prepared a pseudo-labeled training dataset by assigning entirely new class labels. This dataset contained 8.3 M images annotated with 93 K facial class labels, 68 landmarks, and Euler angles using three pretrained networks.In our pseudo-labeling framework, we used an eye blink detection network to align the eye landmarks to make them more robust, which the pretrained network could not achieve. We then employed a selective fitting algorithm to generate high-quality pseudo-labels and validated them using pretrained networks to prevent duplicated face IDs.In our multitask learning framework, we designed a loss that generated a synergy between tasks for pose-invariant face recognition and pose-guided facial landmark detection. In addition, we developed a new face recognition evaluation method for pose-invariant evaluation on the IJB–C [31] dataset.Finally, we constructed a novel pseudo-labeling and multitask learning framework and demonstrated its SOTA or near-SOTA performance, thus illustrating its applicability to not only the three tasks but also the entire field of facial analysis.

## 2. Related Works

### 2.1. Single-Task Learning

#### 2.1.1. Face Recognition

Investigations into face recognition technology using deep learning techniques have mainly focused on refining the datasets and loss functions. The high accuracy in face recognition tasks is largely attributed to the quality of the associated datasets, which have grown immensely in size. Datasets such as CASIA-WebFace [32], VGGFace2 [28], MS-Celeb-1M [33], Megaface [34], and Webface [35] have between thousands and up to 100,000 human classes, and datasets with 4 million human classes have been created recently.

As these large, high-quality datasets become available, the design of practical loss functions for training on these datasets has become crucial. In deep-learning-based methods, the traditional SoftMax loss is based on the Euclidean distance between the embedding features. However, newer methods, such as L-Softmax [36] and A-Softmax [37], focus on angular distances that allow more effective decision boundaries for unseen human classes. Further advances, such as CosFace [38], introduce additional distances in the angular space to distinguish human classes more effectively. ArcFace [4] improves this by adding a margin to the angular distance, thereby further expanding the decision boundary. However, as the datasets have grown, these advanced loss functions have become computationally intensive, particularly in fully connected layers. Partial FC [5] addresses this issue by remembering the center of each human class’s embedding features and employing efficient sampling through multi-GPU setups. It considers only the positives and negatives of each human class, thereby reducing the computational cost of training.

#### 2.1.2. Head Pose Estimation

Head pose estimation is primarily used as a lightweight indicator to determine the user status in areas such as virtual reality/augmented reality (AR) for human–computer interaction, autonomous driving systems, and similar applications.

Algorithms such as POSIT [39] and Perspective-n-Points [40] were used to deduce rotation matrices from landmarks and convert them into pose values. The construction of datasets was challenging because it was necessary to label the Euler angles accurately. Further, a variety of parameters, such as the distance between the camera and subject, the focal length, and other camera parameters, had to be kept consistent, making the process of obtaining the data difficult.

However, owing to the creation of the CMU-MultiPIE [41], BIWI Kinect Head Pose Database [42], and CMU-Panoptic [43] datasets, which are labeled with Euler angles for various poses, deep-learning-based head pose estimation methods such as FSA-Net [44], TriNet [45], and RankPose [46] have shown substantial performance improvements. These datasets have facilitated progress in head pose estimation and facial landmark detection. This task also offers a supplementary tool for the learning of pose-invariant features in face recognition and facial landmark detection, and it employs projection algorithms such as Perspective-n-Points to facilitate mutual estimation.

#### 2.1.3. Facial Landmark Detection

At present, facial landmark detection is being used in various applications, as it involves locating key characteristic points on the face and using them to create 3D face models or produce virtual facial movements by mapping them to the blend shapes of 3D avatars.

Among deep-learning-based methods [6,7,22,47,48,49,50,51], TCDCN [47] represents an early 2D facial landmark detection model using deep learning; it demonstrates superior performance compared with previous nonparametric methods, even with a simplistic CNN layer structure. In contrast, FAN [48] adopts the stacked hourglass network structure to enable 2D and 3D facial landmark detection. Further, the Wing loss [49] has been proposed as a new loss function that focuses on small errors in the loss function to achieve refined learning in landmark detection.

### 2.2. Multitask Learning

In deep learning, multitask learning is often used to improve the performance of a primary task. The MTCNN [19] face detector is a good example of multitask learning in face analysis. It emphasizes the accuracy of face identification in images by simultaneously learning to distinguish between faces and backgrounds and locating five key points to facilitate the accurate determination of the bounding box coordinates. The primary goal of multitask learning is to increase the efficiency of all constituent tasks; nevertheless, the parameters must be carefully assigned to match the network architecture and learning methodology to avoid one-sided biases and achieve a balanced performance improvement.

As an example of good parameter assignment tailored to specific tasks, the all-in-one CNN [27] describes a universal multitask learning model that integrates tasks corresponding to the main areas of face analysis, namely face recognition, face detection, head pose estimation, and facial landmark detection. This method specifies face recognition as the primary network branch from which the subnetwork branches are constructed and subsequently trained for additional tasks. SwinFace [21] introduces a multitask learning approach that achieves SOTA performance while proficiently managing approximately 42 tasks, including face recognition, facial expression, age estimation, and facial attribute estimation; the results demonstrate that this learning method generalizes the network performance for various facial regions of interest (ROI) required in different tasks.

As facial landmark detection and head pose estimation are inter-projectable, a dataset called 300W-LP [22] was created to combine these two tasks into a single dataset, using a face model such as the 3D Morphable Model (3DMM) [52] to enable sophisticated facial landmark detection and head pose estimation. FAN [48], 3DDFA [22,53,54], and SynergyNet [7] leverage 300W-LP [22] to present multitask learning methods that extend beyond facial landmarks and head pose estimation to include 3D face alignment. In particular, SynergyNet [7] focuses on predicting the complete 3D face geometry, including 3D face alignment from 3D face modeling and face orientation estimation using a proposed multitask, multimodal, multi-representation landmark refinement network. This study focuses on 3D landmarks and important face attributes by using their embedded information to guide the learning of the 3D face geometry. SynergyNet demonstrates robustness under various conditions and exhibits SOTA performance in facial landmark detection and head pose estimation.

## 3. Proposed Method

### 3.1. Pseudo-Labeling

In this section, we introduce our pseudo-labeling framework for the integration of three tasks: face recognition (FR), head pose estimation (HPE), and facial landmark detection (FLD).

#### 3.1.1. Dataset Selection Strategy

Table 1 lists the commonly used datasets for each task. We focus on the ratios of the datasets, the number of images per ID, and the total amount of data. We select three datasets as the pseudo-labeled training dataset: two FR datasets, VGGFace2 [28] and MS1MV3 [29], and 300W-LP [22], which is the labeled dataset for HPE and FLD.

BIWI [42] and CMU-Panoptic [43] are mainly used as HPE datasets, and 300 W [30], AFLW [55], COFW [56], and WFLW [57] are mainly used as FLD datasets. However, these HPE and FLD datasets face challenges in pseudo-labeling each other because of an absolute lack of face IDs, insufficient data, or a large domain gap between the datasets. For example, HPE datasets have face images in large poses, whereas FLD datasets have no face image information in large poses, which likely indicates mislabeling. However, HPE and FLD are sufficiently related to allow for mutual perspective projection, and the 300W-LP [22] dataset was developed in another study to contain labels for HPE and FLD tasks. The 300W-LP dataset was created by applying the 3DDFA [22] fitting algorithm through the 3D Face Morphable Model [52] from 300 W; it includes 2D and 3D landmarks and the yaw, pitch, and roll of the pose labels in large pose variants. However, although this dataset was expanded from 3837 images to 61,225 images, it still has issues, such as a lack of face IDs, an inability to detect eye blinks, and the distortion of most images in three dimensions, leading to a lack of diversity in the background domain. To resolve these issues, we attempted to integrate the data from this dataset with the abundant face IDs from the FR datasets and train a multitask network using 300W-LP to attempt pseudo-labeling.

When selecting an FR dataset for pseudo-labeling, as shown in Figure 2, we primarily considered datasets with a balance among the number of images per ID and the total number of IDs. We chose VGGFace2 [28] as it allows sufficient background diversity. However, when training facial recognition networks using VGGFace2, we observed that the small number of face IDs caused a noticeable drop in verification performance in the challenge test dataset, IJB-C [31]. To compensate for this, we considered adding other FR datasets, such as MS1MV3 [29], MegaFace2 [34], and WebFace260M [35], which have several IDs. MegaFace2 was not considered owing to its small number of images per ID, and the training of WebFace260M was computationally expensive. Consequently, we used MS1MV3 for noise and alignment. Finally, we selected two FR datasets, VGGFace2 and MS1MV3. To assign new IDs to the integrated dataset, we prepared two individual networks on each dataset by using the ResNet [58] architecture.

#### 3.1.2. Pseudo-Labeling Landmark and Pose

We devised a pseudo-labeling framework to create an integrated training dataset for multitask learning, as shown in Figure 3. Our pseudo-labeling framework consists of two main steps. Initially, the two FR datasets are subjected to pseudo-labeled landmarks and pose labels. We prepared a pretrained model by using the 300W-LP [22] dataset on the ResNet50 [58] model and concurrently employed Google’s MediaPipe [59] landmark detection framework to minimize bias in our network. One of the issues described earlier is the notable influence of the face detection bounding box area used during training. It affects the generalization of tasks dealing with insufficient face IDs to real-world applications. Therefore, an algorithm that uses multiple detection ROIs for a single image is essential in determining accurate landmark and pose values. Thus, we implemented an algorithm to extract quality pseudo-labels by generating multiple input images from a single image. By applying various mixtures of crops and padding from our data augmentation set, we enabled our pretrained network and MediaPipe [59] to extract landmark and pose values from diverse perspectives. Algorithm 1 describes the sampling and fitting algorithm intended for pseudo-labeling.
**Algorithm 1** Sampling and Selective Fitting Algorithm for Pseudo-Labeling.**Input:** number of total samples *N*, image for sampling **I**, set of augmentations A = $\left\{a_{1},...,a_{N}\right\}$, pretrained network with 300W-LP [22] dataset net, Euclidean distance threshold $t_{d}$.**Output:** Pseudo 68 landmarks and pose values. 1:make a new list D 2:**for** n=1 to *N* **do** 3:    augmented image $I^{\prime }\leftarrow a_{n}\left(I\right)$ 4:    $L_{68},P_{3}\leftarrow net\left(I^{\prime }\right)$ 5:    $I_{eye_{l}}^{\prime },I_{eye_{r}}^{\prime }\leftarrow \mathrm{Crop}\left(I^{\prime },L_{68}\left[37:48\right]\right)$ 6:    $\sigma_{l},\sigma_{r}\leftarrow net_{eye}\left(I^{\prime }eye_{l},I^{\prime }eye_{r}\right)$ 7:    $L_{68}\left[37:48\right]\leftarrow \mathrm{AdjustUpperEyes}(L_{68}\left[37:48\right],\sigma_{l},\sigma_{r}$) 8:    $L_{468}^{med}\leftarrow \mathrm{MediaPipe}\left(I^{\prime }\right)$ 9:    $L_{68}^{med}\leftarrow \mathrm{IndexMatching}\left(L_{468}^{med}\right)$10:    **if** $L_{68}$ is valid landmark **then**11:        append $L_{68},P_{3}$ to D12:    **end if**13:    **if** $L_{68}^{med}$ is valid landmark **then**14:        append $L_{68}^{med},P_{3}$ to D15:    **end if**16:**end for**17:**while** Remaining D>3 **do**18:    $L_{68}^{mean}\leftarrow \mathrm{MeanLandmark}\left(D\right)$19:    $idx\leftarrow \mathrm{CalcMostFarLandmark}(D,L_{68}^{mean})$20:    delete idx in D list21:**end while**22:**if** Remaining D≤3 **then**23:    $L_{68}^{mean}\leftarrow \mathrm{MeanLandmark}\left(D\right)$24:    assign←true25:    **for** $L_{68}^{sample}$ in D **do**26:        **if** Distance$(L_{68}^{sample}$, $L_{68}^{mean})>t_{d}$ **then**27:           assign←false28:        **end if**29:    **end for**30:    **if** assign **then**31:        assign $L_{68}^{mean},P_{3}^{mean}$ pseudo label of I32:    **end if**33:**end if**

In particular, in Algorithm 1, we create multiple image inputs by applying the augmentation set A to each image I in the FR datasets. These inputs are then processed by our pretrained network net and trained using 300W-LP [22] and MediaPipe to extract the landmark vector L and pose vector P, respectively. Representing the landmark vector and the pose vector in Algorithm 1, $L_{k}$ denotes the vector containing the number of *k* landmarks. However, 300W-LP, which we used as the training dataset, rarely includes cases with eye blinks. Consequently, our pretrained model struggles with closed, obstructed, or narrowly opened eyes because it has not been trained using such samples. In contrast, the FR datasets that we chose include a diverse range of cases, and labeling without adjusting for the landmarks corresponding to the eyes would make the resulting pseudo-labeled dataset unable to account for challenging cases such as closed eyes.

Therefore, to incorporate a different level of complexity for eye landmarks compared with 300W-LP [22], we used a CNN-based eye blink detection network, as proposed by Hu et al. [60]. We pretrained an eye blink detection network using the ResNet18 architecture [58] based on the labeling information of cropped eyes and the binary class of opened or closed eyes as refined by Song et al. [61], using the BioID [62], AR [63], and CAS-PEAL [64] datasets.

As shown in Figure 4, we applied the trained eye blink detection network, $net_{eye}$, in Algorithm 1. This network was applied to the eye regions identified by $L_{68}\left[37:48\right]$, corresponding to the 37th to 48th indices of the landmark vector. It extracted $\sigma_{l}$ and $\sigma_{r}$ values ranging from 0 to 1, representing the probability of blinks in the left and right eyes, respectively. In the **AdjustUpperEyes** method in Algorithm 1, these $\sigma_{l}$ and $\sigma_{r}$ values were then multiplied by the difference values, $h_{l1}$, $h_{l2}$, $h_{r1}$, and $h_{r2}$, representing the distances between the upper and lower eyes of each face, to incorporate the weight of the eye part of the landmark. Subsequently, the validity of the extracted landmarks was assessed. Landmarks that were abnormally small compared with the width and height of the input image and those that showed the x-axis coordinates of the landmarks clustered in the center of the image were considered invalid. Only samples validated through this process were incorporated into the list of landmark vectors D.

Meanwhile, as MediaPipe [59] is not trained with 300W-LP [22], we did not apply **AdjustUpperEyes**. Unlike typical landmark detection, MediaPipe outputs 468 landmarks. Therefore, to find the indices of these 468 landmarks that are most similar to those of the 68 landmarks, we used the 300W-LP dataset for the pretraining network. We compared the results from our pretrained network with MediaPipe’s results to extract the 68 landmark indices most analogous to 300W-LP from the 468 landmarks provided by MediaPipe, as illustrated in Figure 5. Similarly, after confirming that the landmark vector extracted via MediaPipe was valid, we added it to the list of landmark vectors D.

For selective fitting, we calculated the average landmark vector from the list of landmark vectors D, excluded the landmark vector with the farthest Euclidean distance from the average landmark vector from the list D, and repeated this step until the length of the list D was three or fewer. These were accepted as pseudo-labels for the image I only when the mean Euclidean distance between $L_{68}^{mean}$ and each of the remaining samples was lower than the Euclidean distance threshold, $t_{d}$. Consequently, the final image would store three or fewer samples’ average landmark and average pose values. The distance threshold $t_{d}$ was established based on the average performance of the AFLW2000-3D [22] test, with a normalized mean square error of 0.05, in the 3D FLD tasks.

#### 3.1.3. Pseudo-Labeling Face Identification

We have described how to label pseudo-landmarks and pseudo-pose labels. Next, we describe a framework for the distribution of face IDs to integrate the dataset in the following Algorithm 2. First, most FR datasets are composed of data on celebrities and other famous individuals. Consequently, some duplicate IDs exist between the FR datasets. To filter out duplicates between the VGGFace2 [28] and MS1MV3 [29] datasets, we calculated the similarity threshold values $t_{MS}$ and $t_{VGG}$ while pretraining the face recognition task with each dataset. During the pretraining process, we used the LFW [65] verification test, commonly employed to evaluate the performance of face recognition tasks, to measure the optimal similarity threshold that maximizes the verification performance. This threshold was then used as the similarity threshold for the detection of duplicates in Algorithm 2. Both angular and Euclidean distances can be used to measure similarity; however, the same distance metric must be employed in the loss function used during pretraining. Since we used the large margin cosine loss [38] function in our proposed method, we employed the angular distance for the cosine similarity in detecting data duplicates. For multitask learning, we filtered out duplicate IDs between VGGFace2 [28] and MS1MV3 [29] and added new IDs to 300W-LP [22] for the pseudo-labeled training dataset.
**Algorithm 2** Identifying Duplicate IDs to Assign New IDs for the Integrated Dataset.**Input:** Pretrained network using VGGFace2 [28] $net_{VGG}$, pretrained network using MS1MV3 [29]$net_{MS}$, VGGFace2 as $D_{VGG}$, MS1MV3 as $D_{MS}$, number of IDs in $D_{VGG}$$l_{VGG}$, number of IDs in $D_{MS}$$l_{MS}$, 300W-LP [22] as $D_{300W}$, similarity threshold $t_{VGG}$ for $net_{VGG}$, and similarity threshold $t_{MS}$ for $net_{MS}$.**Output:** Our pseudo-labeled training dataset $D_{T}$ with self-curated and unduplicated IDs. 1:$E_{VGG}\leftarrow$ Empty list of length $l_{VGG}$+$l_{MS}$ 2:$E_{MS}\leftarrow$ Empty list of length $l_{VGG}$+$l_{MS}$ 3:**for** each ID $i_{VGG}$ in $D_{VGG}$ **do** 4:    $S_{I}\leftarrow \mathrm{set}\;\mathrm{all}\;\mathrm{images}\;\mathrm{in}\;i_{VGG}$ 5:    $E_{VGG}\left[i_{VGG}\right]\leftarrow \mathrm{Mean}\left(net_{VGG}\left(S_{I}\right)\right)$ 6:    $E_{MS}\left[i_{VGG}\right]\leftarrow \mathrm{Mean}\left(net_{MS}\left(S_{I}\right)\right)$ 7:**end for** 8:**for** each ID $i_{MS}$ in $D_{MS}$ **do** 9:    $S_{I}\leftarrow \mathrm{set}\;\mathrm{all}\;\mathrm{images}\;\mathrm{in}\;i_{MS}$10:    $E_{VGG}\left[l_{VGG}+i_{MS}\right]\leftarrow \mathrm{Mean}\left(net_{VGG}\left(S_{I}\right)\right)$11:    $E_{MS}\left[l_{VGG}+i_{MS}\right]\leftarrow \mathrm{Mean}\left(net_{MS}\left(S_{I}\right)\right)$12:**end for**13:${\mathrm{Sim}}_{VGG}\leftarrow \mathrm{ComputeSimilarityList}\left(E_{VGG},t_{VGG}\right)$14:${\mathrm{Sim}}_{MS}\leftarrow \mathrm{ComputeSimilarityList}\left(E_{MS},t_{MS}\right)$15:$U\leftarrow \mathrm{FindDuplicates}({\mathrm{Sim}}_{VGG},{\mathrm{Sim}}_{MS})$16:$D_{T}^{\prime }\leftarrow \mathrm{MergeDatasets}\left(D_{VGG},D_{MS},U\right)$17:$D_{300W}^{\prime }\leftarrow \mathrm{SemiIdentify}\left(D_{300W}\right)$18:$U^{\prime }\leftarrow \mathrm{repeat}\;\mathrm{step}\;1\;\mathrm{to}\;15\;\mathrm{using}\;D_{T}^{\prime },D_{300W}^{\prime }$19:$D_{T}\leftarrow \mathrm{MergeDatasets}\left(D_{T}^{\prime },D_{300W}^{\prime },U^{\prime }\right)$

As we integrated VGGFace2 [28] and MS1MV3 [29] as $D_{T}^{\prime }$, we needed to assign new IDs to 300W-LP [22], which does not have any existing IDs. In particular, 300W-LP is a dataset that distorts each image of 300W [30], which holds approximately 3800 images, into 10 to 20 images, thus augmenting the dataset. Consequently, we assumed that each of the approximately 3800 images possessed a unique ID. We assigned arbitrary IDs to 61,225 images as $D_{300W}^{\prime }$ and checked for duplicate IDs at $D_{T}^{\prime }$. We reviewed 92 internal duplicate IDs $U^{\prime }$ and directed them to an integrated dataset of VGGFace2 and MS1MV3. We confirmed the absence of duplicate IDs and prepared the pseudo-labeled training dataset labeled with 8,271,075 images and 93,094 IDs, along with pseudo-landmark and pseudo-pose values. Figure 6 illustrates the constructed training dataset.

### 3.2. Multitask Learning

In this section, we discuss how to train the three tasks using the integrated training dataset discussed in the previous section. In particular, because the pseudo-labeled training dataset used pseudo-labeling rather than exact labeling, outliers inevitably existed, and we needed to find a learning technique to reduce these imperfections. The multitask learning of different facial analysis tasks also requires coordination on the optimal face regions required by each task. For example, for facial landmark detection, optimal learning results are achieved when the face region includes a significant portion of the head, chin, and neck. In contrast, face recognition can achieve good results when the background is minimized and major facial features such as the eyes, nose, and mouth are correctly aligned in a cropped face area. To address this performance trade-off based on the face area, we considered the required bounding box area of the face to determine the maximum area that could be used without degrading the performance of the task, and we actively used data augmentation methods to cut out arbitrary face areas between the maximum and minimum areas and use them for training.

#### 3.2.1. Network Architecture

To show that our proposed learning method is feasible, we chose ResNet [58], a network structure commonly used in face-recognition-based networks, as a baseline and used the ResNet50 structure for our pretrained model to make the comparison as fair as possible. Furthermore, we used the modified IResNet50 [66], which is used in recent face-recognition-based methods such as ArcFace [4] and Partial FC [5], as the backbone of the multitask learning network. IResNet has no critical differences from ResNet, but we chose and modified the IResNet structure because it allowed us to appropriately modify the details of the normalization and activation layers required by each task in face recognition, facial landmark detection, and head pose estimation. While tuning the network architecture for each task, we also considered maximizing the shared parameters to minimize the inference time loss. To do this, we referenced several face analysis tasks [7,44,67,68] and finally split the shared and independent parameters at a 3:1 ratio for each task in the baseline network. The modified IResNet50 consists of four layers, where the first three layers are shared layers and the last layer is divided into sublayers customized for each task, as shown in Figure 7.

#### 3.2.2. Multitask Loss Functions

To ensure balanced learning between tasks and avoid convergence to a specific task, weights must be assigned to each task, as given in Equation (Equation 1): (1)$$L_{total}=L_{FR}+\lambda_{1}L_{FLD}+\lambda_{2}L_{HPE},$$
where $L_{FR}$, $L_{FLD}$, and $L_{HPE}$ denote the loss functions of face recognition, facial landmark detection, and head pose estimation, respectively. We set the base criterion at 1 for $L_{FR}$ and applied the learning weights $\lambda_{1}$ and $\lambda_{2}$ to $L_{FLD}$ and $L_{HPE}$, respectively. These weights were adjusted based on the sum of the values represented by the final output of each task. We fine-tuned each task’s contribution to the overall multitask framework by choosing $\lambda_{1}$ to be 100 and $\lambda_{2}$ to be 10 to provide an appropriate learning ratio based on the final convergence of each task’s loss function. We created our multitask loss function, described in Equation (Equation 1), by computing the total loss value $L_{total}$ by summing all weighted losses from each task.

The loss function for the head pose estimation task is given in Equation (Equation 2): (2)$$L_{HPE}=\frac{1}{N_{P}}\sum_{i=1}^{N_{P}}\left|P_{PRED}^{i}-P_{GT}^{i}\right|,$$
where $N_{P}$ is the number of elements (=3), representing yaw, pitch, and roll, and P is the pose vector including these three Euler angle values. It is calculated by computing the mean absolute error between the predicted pose vector $P_{PRED}$ and the ground truth pose vector $P_{GT}$. The head pose tends to be less influenced by the face region than the other two tasks and achieves convergence in the shortest time during training. Therefore, we designed the losses for the remaining two tasks to complement each other by assigning learning weights through the pose values to the other two tasks.

We prepared two major loss functions for the loss function of facial landmark detection $L_{FLD}$, as presented in Equation (Equation 3): (3)$$L_{FLD}=L_{wing}+\lambda_{P}L_{MSE},$$
where one is the Wing loss $L_{wing}$[49] and the other is the mean square error (MSE) loss multiplied by $\lambda_{P}$. $\lambda_{P}$ is a regularization term that controls the convergence between the Wing loss and the MSE loss. The method for the determination of $\lambda_{P}$ and its effects on the convergence of the losses are described in Equation (Equation 6) through (Equation 9). The Wing loss $L_{wing}$ is given by Equation (Equation 4): (4)$$L_{wing}=\left\{\begin{array}{cc} w\ln(1+|x|/\epsilon ) & \;\mathrm{if}\;|x|<w\\ |x|-C & \;\mathrm{otherwise}\; \end{array}\right.,$$
where *w* denotes the non-negative value to set the range of the nonlinear part, ϵ limits the curvature region, and C=w−wln(1+w/ϵ) is the smoothness between the linear and nonlinear parts. L denotes the landmark vector and *x* denotes the L1 norm of the Euclidean distance between the predicted landmark vector $L_{PRED}$ and the ground truth landmark vector $L_{GT}$.

Equation (Equation 3) also contains the loss of the mean square error, commonly used for regression tasks, as given in Equation (Equation 5): (5)$$L_{MSE}=\frac{1}{N_{L}}\sum_{i=1}^{N_{L}}{\left(L_{PRED}^{i}-L_{GT}^{i}\right)}^{2}\;,$$
where $N_{L}$ represents the total number of landmarks, $L_{PRED}^{i}$ is the *i*-th landmark coordinate of the predicted landmark vector, and $L_{GT}^{i}$ is the *i*-th landmark coordinate of the ground truth landmark vector. Our MSE loss, combined with $\lambda_{P}$ in Equation (Equation 3), focuses more on large errors to emphasize the importance of accurate landmark prediction in the presence of pose variations, making it a counterpart to the Wing loss [49] in our facial landmark detection task.

To compute $\lambda_{P}$, we need to derive the projected landmark vector $L_{P}$ from the mean 3D landmark $L_{3D}$ used to calculate the samples in 300W-LP [22] by using the ground truth pose values of the pitch, yaw, and roll. The mean 3D landmark vector, denoted as $L_{3D}$, comprises 68 landmark points, each represented by three-dimensional Euclidean space coordinate vectors x, y, and z. This vector is derived from the average landmark vector of samples in the frontal view of 300W-LP with pitch, yaw, and roll values close to zero. This mean 3D landmark is then multiplied by the projection matrix to locate the projected landmark vector given by
(6)$$L_{3D}={\left[\begin{array}{ccc} x & y & z \end{array}\right]}^{T},$$

Following the derivation of the mean 3D landmark, the rotated point $L_{3D}^{\prime }$ was calculated. This point is a product of the rotation matrix R, taken from Rodrigues’ formula [69], and the original 3D landmark, $L_{3D}$, resulting in new coordinates $x^{\prime }$, $y^{\prime }$, and $z^{\prime }$. The interaction between these components is summarized in Equation (Equation 7): (7)$$L_{3D}^{\prime }=R\cdot L_{3D}={\left[\begin{array}{ccc} x^{\prime } & y^{\prime } & z^{\prime } \end{array}\right]}^{T},$$
where the projected 2D landmark vector $L_{P}=\left(x_{P},y_{P}\right)$ can be obtained using perspective projection to the rotated point $L_{3D}^{\prime }$ in Equation (Equation 8): (8)$$L_{P}=\left[\begin{array}{c} x_{P}\\ y_{P} \end{array}\right]=\left[\begin{array}{ccc} f_{x} & 0 & c_{x}\\ 0 & f_{y} & c_{y} \end{array}\right]\cdot L_{3D}^{\prime },$$
where $f_{x},f_{y}$ are the focal lengths; $c_{x},c_{y}$ are the principal points; and $x_{P}$, $y_{P}$ are 2D projected landmark coordinate vectors. These values can be determined based on the input shapes or assigned arbitrarily. $\lambda_{P}$ is calculated using Equation (Equation 9): (9)$$\lambda_{P}=\mathrm{mean}\left(\frac{L_{P}}{\left|\right|L_{P}\left|\right|}-\frac{L_{PRED}}{\left|\right|L_{PRED}\left|\right|}\right),$$
where $\lambda_{P}$ is the mean of the Euclidean distance between the norms of the predicted landmark vector $L_{PRED}$, projected landmark vector $L_{P}$, and visualization representation shown in Figure 8. The primary intent behind employing $\lambda_{P}$ is not necessarily to enhance the accuracy but to facilitate the initial learning of the landmarks. Given the possibility that the pseudo-labeling of the ground truth is erroneous, $\lambda_{P}$ also serves as a counterpart to address potential inaccuracies, acting as a stabilizing component in our multitask loss function.

With regard to the loss function for face recognition, we acknowledge the effectiveness of representing the embedding features in the angular domain through the angular margin, which has been proven to enhance the performance significantly. Consequently, we adopted the marginal cosine loss function method, conceptualized in approaches such as CosFace [38], SphereFace [37], and ArcFace [4], due to the novel representation in the angular domain.

Building on the foundational work on the large margin cosine loss [38], we focus on the angular region specific to each ID. To achieve this, we apply the pose ratio as a margin, allowing a greater emphasis on the angular region of each ID. As shown in Figure 9, the allocation of feature regions according to the pose values of each ID tends to drift away from the center of the feature region that the ID holds as the pose values increase. To address this concern and assign feature regions unaffected by the pose, we set margins to allow more challenging pose values closer to the center of their respective ID. $L_{FR}$ represents the large margin cosine loss originally proposed in [38]. However, in this study, we adapted it to incorporate the additional variance associated with pose values in the margin, as given in Equation (Equation 10).
(10)$$\begin{array}{cc} L_{FR} & =L_{\mathrm{LPMC}}\left(\theta ,y\right)\\ \; & =\frac{1}{N_{F}}\sum_{i}-log\frac{e^{s\cdot cos(\theta_{y_{i}}^{i}-m_{p})}}{e^{s\cdot cos(\theta_{y_{i}}^{i}-m_{p})}+\sum_{j\neq y_{i}}e^{s\cdot cos\theta_{j}^{i}}}, \end{array}$$
subject to
(11)$$cos\left(\theta_{j}^{i}\right)=W_{j}^{T},x_{i},$$
where $N_{F}$ is the number of training samples, and $x_{i}$ is the *i*-th normalized feature vector with a corresponding ground truth class of $y_{i}$. $W_{j}$ is the normalized weight vector of the *j*-th class, and $\theta_{j}$ is the angle between $W_{j}$ and $x_{j}$, as conceptualized in CosFace [38]. A novel margin, $m_{p}=m_{1}+m_{2}*\gamma_{P}$, is proposed, where $\gamma_{P}$ is the average of the three poses—yaw, pitch, and roll—summed and then divided by 180 to ensure a value ranging between 0 and 1.

When using our proposed large pose margin cosine loss, we assign more margin values as a weighted indication that the class is correct as the head pose value of the input face image increases. This enables the network to achieve tighter class clustering despite the diverse variations in head pose. Incorporating pose-invariant feature regions provides a means to effectively address and accommodate more difficult pose values, creating an improved and reliable face recognition model.

## 4. Experimental Results

In this section, we compare the results against those of various SOTA methods for each task and provide further understanding through an ablation study. We prepared three different network backbones, namely ResNet50 for our pretrained network, IResNet50 for our single-task learning (STL), and IResNet50-Multi for our multitask learning (MTL) model. As described in Section 3.2, the first three layers of each ResNet structure are shared layers, with the remaining layer allocated to each respective task. Our pretrained model, ResNet50 [58], used images of size 128 × 128, whereas our STL and MTL models using IResNet50 [66] were trained with images of size 112 × 112. These networks used $\lambda_{1}=100$ and $\lambda_{2}=10$ as parameters for the integrated loss $L_{total}$ in Equation Equation 1, with C=10 and ϵ=2 as hyperparameters for the Wing loss [49] in Equation (Equation 4). Finally, for $L_{FR}$ in Equation (Equation 10), the hyperparameters s=64, m1=0.3, and $m_{2}=0.7$ were used, which are generally used in face recognition tasks. We started learning with a learning rate of 0.01, multiplying by 0.9 at every epoch, and trained on 8.3 M data per epoch for 20 epochs. The training was conducted on a computer with NVIDIA’s RTX A6000. Each epoch took approximately 3 h when training in FP16 and 6 h in FP32, resulting in a total training time of 60−120 h.

### 4.1. Face Recognition

#### 4.1.1. Test Dataset

We prepared four datasets for the face recognition task, namely LFW [65], CFP-FP [70], AgeDB [71], and IJB-C [31], which are commonly used for performance comparisons, as shown in Table 2. LFW, CFP-FP, and AgeDB contain 13 K, 7 K, and 16.5 K images, respectively, and evaluations were performed on 6 K, 7 K, and 6 K verification pairs, respectively. These datasets were used to monitor the learning process of face recognition during multitask training. IJB-C has 148.8 K images exhibiting large pose and age variations. Although multiple testing methods, including verification and identification, are available for this dataset, only 1:1 verification was performed in this study. With 15M pairs in this test dataset’s protocol, it is suitable for evaluating the true acceptance rate (TAR) against the false acceptance rate (FAR), and many face recognition methods conduct TAR@FAR evaluations ranging from FAR = $1\times 10^{-1}$ to FAR = $1\times 10^{-6}$. This study conducted evaluations across the entire protocol for IJB-C 1:1 verification. Furthermore, we focused on our pose-invariant learning method using our loss function and categorized IJB-C cases based on the pose values into soft, medium, and hard, as measured internally.

#### 4.1.2. Experimental Results

Table 3 presents the accuracies obtained in the LFW [65], CFP-FP [70], and AgeDB [71] tests. Our pretrained models achieved 99.33%, 92.38%, and 94.20%, respectively, using VGGFace2 [28], and 99.85%, 98.70%, and 98.11%, respectively, using the MS1MV3 [29] dataset. Furthermore, our STL model, trained using the pseudo-labeled training dataset (PLTD) that we developed through pseudo-labeling and combining MS1MV3, VGGFace2, and the 300W-LP dataset, achieved 99.72%, 97.73%, and 96.9%, respectively, in the face recognition task. Our MTL model, incorporating the large pose margin cosine loss, achieved 99.70%, 97.26%, and 96.53%, respectively. Compared with the SOTA models, while most comparative models use ResNet100 [58] or IResNet100 [66] as their backbones, our model deploys ResNet50 [58] and IResNet50 [66]. Despite such differences, our best MTL model exhibited competitive performance in IJB-C [31], achieving 94.17% at TAR@FAR = $1\times 10^{-4}$.

#### 4.1.3. Discussion

In Section 3.1, we reassigned face IDs during dataset integration through pseudo-labeling. Duplication may have been misjudged as an error in some instances; these are referred to as outliers in our context. However, the verification performance summarized in Table 3 is comparable to, if not synonymous with, that of other SOTA algorithms. It is inferred that the test performance might have partially diminished because we did not crop the faces tightly or align them during the training stage, as done by other face recognition methods while using the dataset. Additionally, despite training with ResNet100, similar to SOTA configurations, our performance metrics were identical to those achieved with ResNet50.

These results indicate that filtering and learning from them is feasible even if the pseudo-labeled dataset incorporates some outliers. It also exhibits the potential to effortlessly integrate various facial analysis tasks beyond the proposed methods, highlighting the versatility and adaptability of our approach despite the anomalies and discrepancies within the data. In addition, we compared the inference time with that of similar works under identical conditions and conducted a frame-per-second (fps) comparison for the ResNet50 and IResNet50 models used in the comparison works. Unlike the recent trend of using arrays of multiple GPUs to train face recognition methods, our fps comparison was conducted using only a single GPU. Despite performing multitask computations, our proposed method performed well in terms of fps.

Next, in Table 4, we present the experimental results regarding the large pose margin cosine loss designed for our MTL and introduce a new IJB-C [31] test method for the evaluation of pose invariance. The IJB-C 1:1 verification method stores the average of all embedding features of the templates corresponding to each ID within the dataset. Then, it conducts a TAR@FAR performance evaluation by comparing the templates over 15 million pairs. Here, while we store the average of each template’s embedding feature in the gallery similarly, for the probe, we detect the yaw, the z-axis rotation angle of each face in each template, through our MTL model. Based on the yaw values, we name the images within $0^{\circ }$ to $30^{\circ }$ as soft cases, $30^{\circ }$ to $60^{\circ }$ as medium cases, and $60^{\circ }$ to $90^{\circ }$ as hard cases. We save the average of the images corresponding to each case in each template as an embedding feature to use as the probe. Ultimately, among 15,658,489 pairs in the IJB-C 1:1 verification method, 15,070,066 pairs have a soft case, 11,410,991 pairs have a medium case, and 5,926,383 pairs have a hard case, with the possibility of duplicates owing to the presence of all three cases within one template.

In Table 4, for a fair comparison, only our single-task and multitask learning models are used for the evaluation. Both applied the same learning rate and hyperparameters and concluded the training at the same epoch for comparison. If we compare IJB-C [31] TAR@FAR = $1\times 10^{-5}$ and $1\times 10^{-4}$, the STL model performs better in all cases; however, our model performs better in medium and hard cases. Accordingly, we observed that the large pose margin cosine loss that we applied could represent more accurate embedding features for such large pose cases. However, compared to the soft cases, many aspects still need improvement. We anticipate that we will attain performance comparable to that of other SOTA methods by repeatedly refining our pseudo-labeled training dataset throughout our entire framework.

### 4.2. Head Pose Estimation

#### 4.2.1. Test Dataset

For head pose estimation, we used two test datasets. One was AFLW2000-3D [22], which was relabeled from AFLW [55] using the 300W-LP method [22]. While the original and re-annotated versions exist, we only used the original version for testing because the re-annotated version has all faces tightly cropped. This dataset comprises 2000 images, including Euler angles and 68 facial landmarks. The other dataset was BIWI [42], where the evaluation was performed on 15,678 images. In addition, because the Euler angle variation of 300W-LP used for pseudo-labeling was between $-99^{\circ }$ and $99^{\circ }$, only test samples within the Euler angle were used for AFLW2000-3D and BIWI.

#### 4.2.2. Experimental Results

The evaluation method used for the test employed the MAE of the Euler angle, as in other methods, to conduct the evaluation. Table 5 presents the comparative results for AFLW2000-3D [22]. Our multitask learning model achieved SOTA performance with an AFLW2000-3D MAE score of 3.32. Furthermore, as observed from Table 6, it achieved state-of-the-art results with an MAE score of 3.54.

#### 4.2.3. Discussion

We achieved SOTA performance on both AFLW2000-3D [22] and BIWI [42], the representative test datasets for head pose estimation. This demonstrates that our pseudo-labeled training dataset allowed our multitask learning framework to achieve positive performance improvements in each task. By adding a large amount of face IDs and diverse background noise to address the lack of diversity in existing datasets, as identified in Section 1, we achieved higher accuracy for unseen images in the head pose estimation task.

### 4.3. Facial Landmark Detection

#### 4.3.1. Test Dataset

We evaluated our model on the AFLW2000-3D [22] dataset, which was also used for head pose estimation. We measured the performance of our model using the normalized mean square error (NMSE):(12)$$\mathrm{NMSE}=\frac{1}{N_{L}}\sum_{i=1}^{N_{L}}\frac{{\left(L_{PRED}^{i}-L_{GT}^{i}\right)}^{2}}{d}\;,$$
where $N_{L}$ denotes the number of landmarks in landmark vector L, $L_{PRED}^{i}$ is the *i*-th landmark coordinate of the predicted landmark vector, $L_{GT}^{i}$ is the *i*-th landmark coordinate of the ground truth landmark vector, and *d* is the size of the facial bounding box for the normalized mean square error.

#### 4.3.2. Experimental Results

Table 7 shows a comparison of the performance of various methods and the proposed model in terms of the NMSE. Evaluations were carried out in three different pose ranges based on the yaw rotation angle of the face: from $0^{\circ }$ to $30^{\circ }$, from $30^{\circ }$ to $60^{\circ }$, and from $60^{\circ }$ to $90^{\circ }$. The mean values were calculated by averaging the results within each pose range. Our MTL model demonstrated NMSE scores of 2.62, 3.45, and 4.51 in the respective pose variations. Although our model did not achieve SOTA performance in all variations, it showed superior performance within the lower range of pose variations from $0^{\circ }$ to $30^{\circ }$, exceeding the results of SynergyNet [7].

#### 4.3.3. Discussion

Based on the ground truth, our model achieved its best performance in the range of yaw values from $0^{\circ }$ to $30^{\circ }$. Unfortunately, it did not perform as well in other angle ranges. This result is probably due to the characteristics of the pseudo-labeled training dataset that we created. As discussed previously, pseudo-labeling allows outliers to be included in the dataset, and these outliers are likely to be included as the pose variation of the face images used for pseudo-labeling increases. The pretrained model that we used for pseudo-labeling in Section 3.1 was trained using 300W-LP [22], which has a higher degree of distortion in images with a large pose variation and a nonuniform definition of invisible landmark points, and we suspect that these features were reflected in the pseudo-labeling. However, despite these challenges, our MTL model achieved near-SOTA performance, as shown in Table 7. This notable result highlights the robustness and adaptability of our approach in handling multiple tasks and varying conditions effectively, especially considering that the model achieves this while sharing the network parameters with other tasks for multitask learning.

### 4.4. Ablation Study

In this section, we analyze through additional experiments whether multitask learning affects the performance improvement, the influence of the hyperparameters $\lambda_{1}$ and $\lambda_{2}$ of the loss function designed in Section 3.2, and the influence of the regularization term $\lambda_{P}$.

#### 4.4.1. Influence of Multitask Learning

To demonstrate the effectiveness and stability of our proposed multitask learning framework through pseudo-labeling, we conducted experiments, as shown in Table 8. Table 8 shows that multitask learning configurations can outperform single-task learning models. All models were trained with a pseudo-labeled training dataset (PLTD). While the STL models did not perform optimally due to mislabeled outliers in the training dataset, the MTL models were trained with an optimal network parameter distribution for each task.

#### 4.4.2. Influence of Regularization Term $\lambda_{P}$

The $\lambda_{P}$ described in Equation (Equation 3) is a regularization term that we adopted because the proportion of large pose variation data in the large datasets was relatively small. If we did not adopt $\lambda_{P}$, we would have been required to learn by excluding the difficult data in the large datasets, by the principle of convergence to the mean in deep learning, which affects the convergence of head pose estimation, as shown in Table 9. A $\lambda_{P}$ value of 0 indicates that no regularization strength is used, a $\lambda_{P}$ value of 1 indicates fixed strength for the facial landmark detection loss function, and a $\lambda_{P}$ value of α indicates regularization strength that is dynamically computed during training iterations.

#### 4.4.3. Influence of Hyperparameters $\lambda_{1}$ and $\lambda_{2}$

In multitask learning, adjusting the learning weights for the loss function designed for each task is essential for the optimization process. In this paper, we take the loss function for face recognition as the baseline in the unified loss function designed in Equation (Equation 1) and set the hyperparameter coefficients of $\lambda_{1}$ and $\lambda_{2}$ to the other tasks, facial landmark detection and head pose estimation, respectively. Table 10 shows the results of the experiments conducted to find the optimal hyperparameters. In detail, the loss convergence ratio is the ratio of $L_{FR}$, $\lambda_{1}L_{FLD}$, and $\lambda_{2}L_{FLD}$ in Equation (Equation 1) based on the loss value of each task when the learning converges to the maximum. The results show that the convergence ratio should be distributed to a certain extent to converge to more optimal multitask learning. Learning by increasing the hyperparameter coefficients for the remaining two tasks is more effective in our proposed multitask learning framework than learning by expanding the hyperparameter coefficients for face recognition, which constitutes most of the pseudo-labeled training dataset.

### 4.5. Visualization

In this section, we compare our model with SynergyNet [7], a SOTA method that implements multitask learning for facial landmark detection and head pose estimation to produce visualization results. The results of AFLW2000-3D [22] are visualized in Figure 10. Both models could identify the landmarks and Euler angles with a high degree of accuracy; however, unsuccessful detection occurred in some cases with a large pose variation.

Figure 11 shows the performance of our model and SynergyNet [7] when applied to different ROIs of the face in images from IJB-C [31]. It demonstrates that our trained model can accurately detect landmarks in different face ROIs by learning diverse face ID information and background noises in the training dataset through pseudo-labeling. In contrast, SynergyNet [7] requires the precise detection of the face area through a detector owing to its insufficient training in various situations. The results in Figure 11 suggest that our model can be used with various face detectors. Our training framework provides more reliable performance for face recognition, head pose estimation, and facial landmark detection, making it more suitable for practical applications.

## 5. Conclusions

In this study, we addressed the problem of tasks with insufficient facial analysis performing well on test datasets but performing poorly when applied in the real world. To solve this problem, we used a dataset of face recognition tasks with a large number of face IDs and performed data enlargement through pseudo-labeling to create a multitask training dataset. We constructed a multitask learning method and achieved SOTA or near-SOTA performance. By using multitask learning, we configured a pose-guided loss function to achieve pose-invariant performance in face recognition and accurate learning in facial landmark detection, even when using a pseudo-labeled dataset containing outliers. We believe that our framework can be applied to various facial analyses.

## Figures and Tables

**Figure 1 sensors-24-03212-f001:**
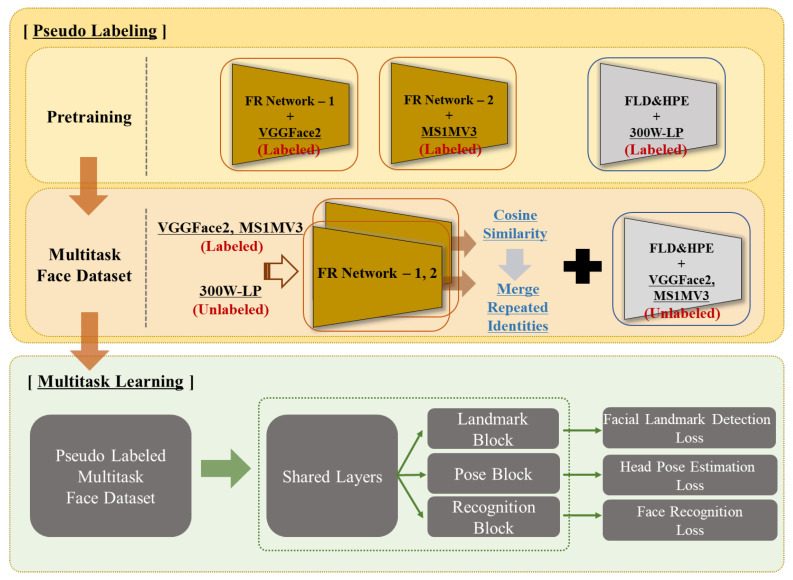
Overview of the integrated framework, beginning with pretraining on individual tasks, followed by pseudo-labeling to generate the pseudo-training dataset, which is then used in our multitask learning. The direction of all arrows indicates the order of application of the framework.

**Figure 2 sensors-24-03212-f002:**
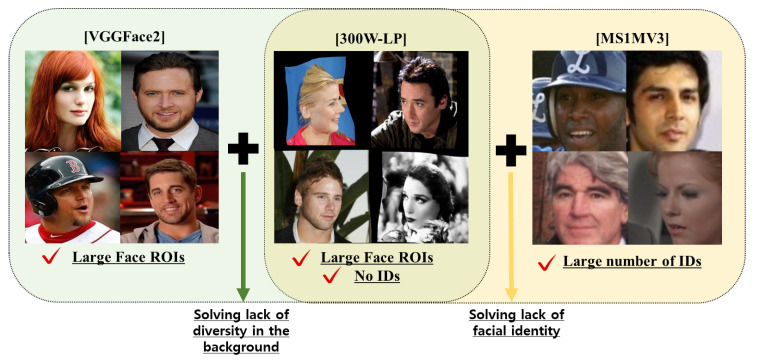
Selecting the face recognition datasets to address the lack of face IDs and diversity in the background in the 300W-LP [22] dataset. Each arrow indicates why we use the VGGFace2 [28] and MS1MV3 [29] datasets.

**Figure 3 sensors-24-03212-f003:**
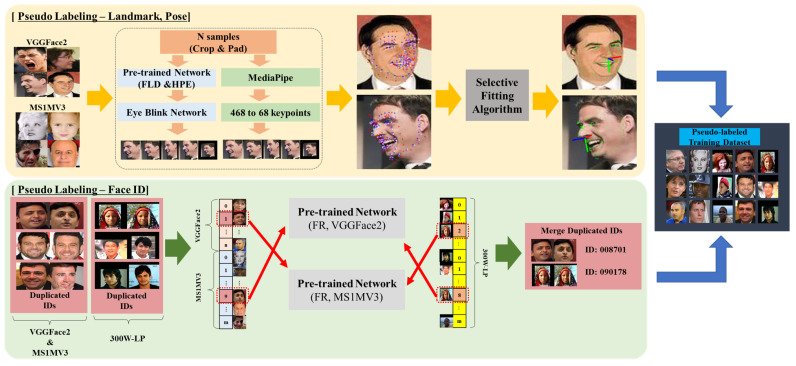
Overview of pseudo-labeled training dataset generation for multitask learning. The upper part illustrates the application of the pseudo-labeling framework of landmarks and poses to the face recognition datasets. The bottom part describes the method of pseudo-labeling face IDs for our training datasets, which involves duplicate ID verification and assigns new face ID labels to the VGGFace2 [28], MS1MV3 [29], and 300W-LP [22] datasets. The direction of all arrows indicates the order of application of the framework.

**Figure 4 sensors-24-03212-f004:**
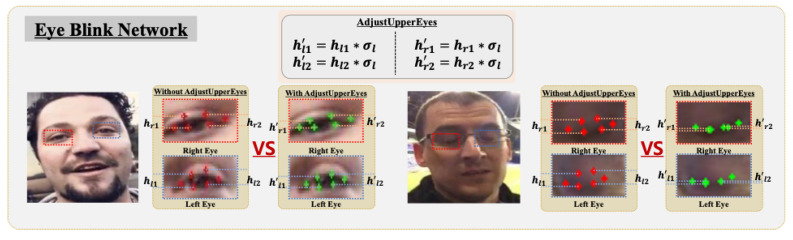
Illustration of handling of closed or partially closed eyes using an eye blink network to address the limitations of the FLD task. We train the ResNet18 model on cropped eye blink datasets to predict the probability of eyes being open or closed. The red dots represent the eye’s landmarks before adjustment and the green dots represent the landmarks after adjustment.

**Figure 5 sensors-24-03212-f005:**
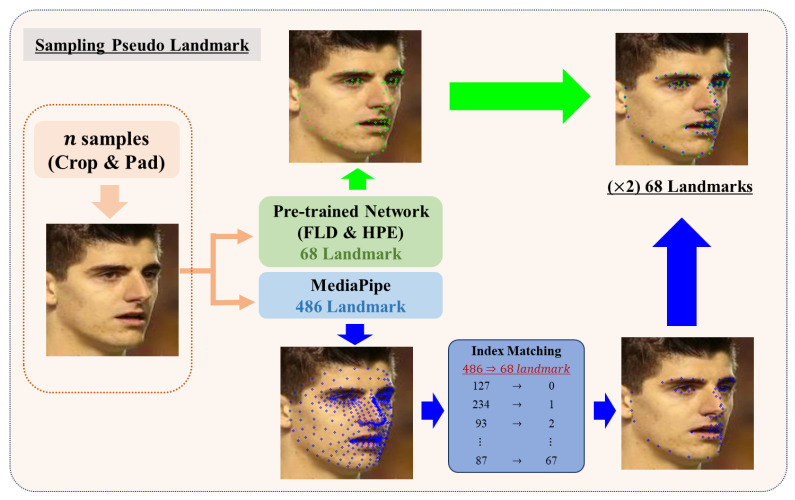
Illustration of sampling of a pseudo-landmark label from Algorithm 1. We used our pretrained network and Google’s MediaPipe [59] to generate a diverse set of pseudo-landmark samples. We applied index matching to select 68 landmarks from the 468 landmarks output by MediaPipe. The green arrows and dots show the flow and output of the pretrained network on the Algorithm 1, while the blue arrows and dots show the flow and output of the MediaPipe.

**Figure 6 sensors-24-03212-f006:**
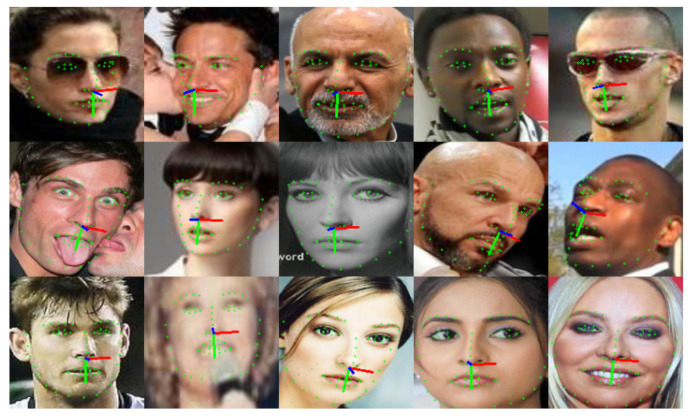
Visualization of our pseudo-labeled training dataset. It contains 8.3 M images and 93 K face IDs with pose and landmark label annotations.

**Figure 7 sensors-24-03212-f007:**
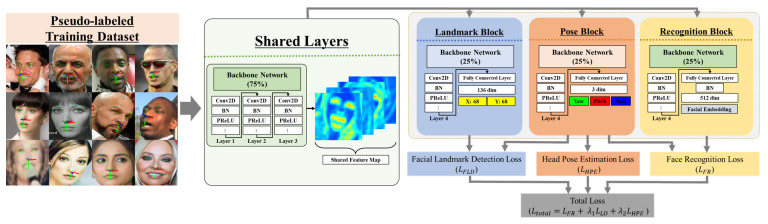
Illustration of our multitask learning network based on the modified IResNet [66], consisting of four layers. The first three layers serve as shared layers across all tasks, and the last layer is duplicated as a sublayer for each task. The direction of all arrows indicates the order of application of our network layers.

**Figure 8 sensors-24-03212-f008:**
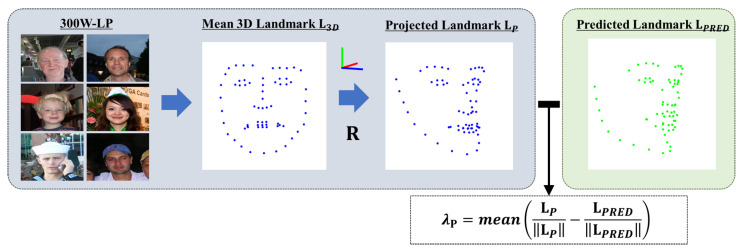
Illustration of the calculation of $\lambda_{P}$, depicting the computation of the mean Euclidean distance between the normalized projected landmark vector $L_{P}$ and the normalized predicted landmark vector $L_{PRED}$.

**Figure 9 sensors-24-03212-f009:**
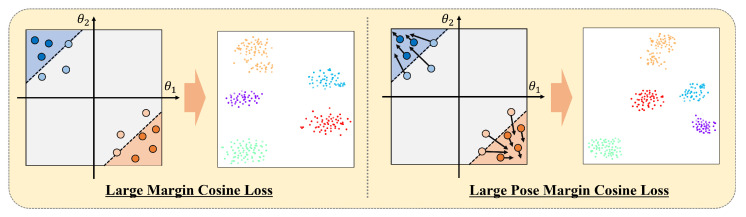
Visualization of loss functions of CosFace [38] and ours with t-sne results. We applied a new margin $m_{p}$ to the loss of CosFace for pose-invariant learning.

**Figure 10 sensors-24-03212-f010:**
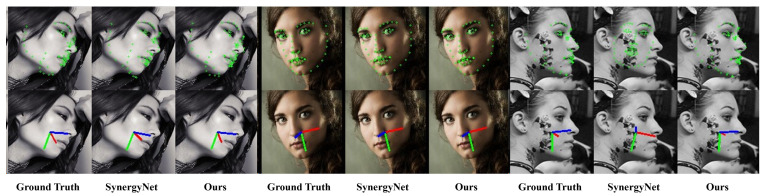
Visualization of ground truth, SynergyNet [7], and our model for HPE and FLD on AFLW2000-3D [22].

**Figure 11 sensors-24-03212-f011:**
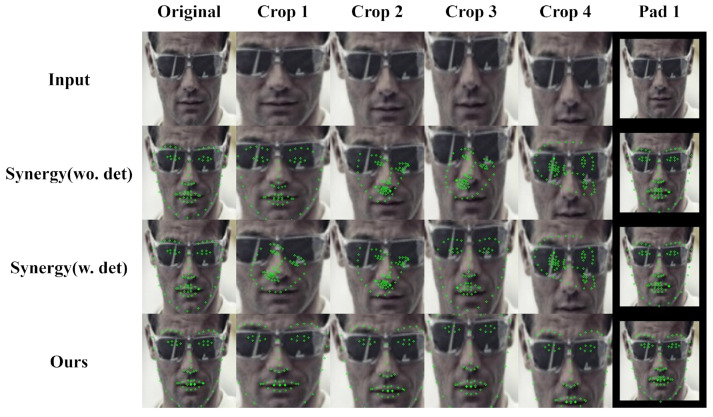
Visualized results of the FLD task between SynergyNet [7] and our model under various face ROIs. The original SynergyNet uses its face detector before facial landmark detection; we compared our model with SynergyNet with and without the detector.

**Table 1 sensors-24-03212-t001:** Datasets for face recognition, head pose estimation, and facial landmark detection.

Dataset	# Images	# IDs	Image/ID	# Landmarks	Pose
CASIA-WebFace [32]	0.5 M	10 K	47	-	
VGGFace2 [28]	3.3 M	9 K	363	-	
MS1MV3 [29]	5.2 M	93 K	56	-	
MegaFace2 [34]	4.7 M	0.6 M	7	-	
WebFace260M [35]	260 M	4 M	21	-	
300W [30]	3.8 K	-	-	68	
AFLW [55]	20 K	-	-	21	
COFW [56]	1.3 K	-	-	29	
WFLW [57]	7.5 K	-	-	98	
BIWI [42]	15 K	-	-	-	✓
CMU-Panoptic [43]	1.3 M	-	-	-	✓
300W-LP [22]	61 K	-	-	68	✓

**Table 2 sensors-24-03212-t002:** Face recognition test datasets used for comparison of verification accuracy.

Dataset	# of ID	# of Images	# of Verification Pairs
LFW [65]	5.7 K	13 K	6 K
CFP-FP [70]	500	7 K	7 K
AgeDB [71]	568	16.5 K	6 K
IJB-C [31]	3.5 K	148.8 K	15 M

**Table 3 sensors-24-03212-t003:** Evaluation results (%) for face recognition datasets LFW, CFP-FP, AgeDB, and IJB-C. Using TAR@FAR = $1\times 10^{-4}$ for IJB-C. **Bold** denotes the best performance, and higher values reflect better results.

Method	Dataset	LFW	CFP-FP	AgeDB	IJB-C	FPS
SphereFace [37]	CASIA	99.42	-	-	-	-
CosFace [38]	CASIA	99.73	-	-	-	-
SphereFace (Re-Imp)	MS1MV2	99.67	98.46	98.17	91.77	-
CosFace (Re-Imp)	MS1MV2	99.78	98.26	98.17	95.56	-
CircleLoss [72]	MS1M	99.73	96.02	-	93.95	-
CurricularFace [73]	MS1MV2	99.80	98.37	98.32	96.10	329.8
ArcFace [4]	MS1MV2	99.82	98.49	98.05	96.03	331.5
ArcFace [4]	MS1MV3	99.83	**99.03**	**98.17**	96.5	331.5
MagFace [74]	MS1MV2	99.83	98.46	98.17	95.97	254.3
Partial FC [5]	MS1MV3	99.85	98.7	98.11	96.08	255.1
SwinFace [21]	MS1MV2	**99.87**	98.60	98.15	**96.73**	70.3
Ours (pretrained)—ResNet50	VGGFace2	99.33	92.38	94.20	88.05	321.7
Ours (pretrained)—ResNet50	MS1MV3	99.85	98.70	98.11	96.08	321.7
Ours (STL)—IResNet50	PLTD	99.72	97.73	96.90	94.05	253.2
Ours (MTL)—IResNet50-Multi	PLTD	99.70	97.26	96.53	**94.17**	221.2
Ours (MTL)—IResNet100-Multi	PLTD	99.68	97.22	96.29	94.09	112.0

**Table 4 sensors-24-03212-t004:** Evaluation results (%) for our single-task learning (STL) and multitask learning (MTL) models on the IJB-C test dataset across all, soft, medium, and hard cases, with thresholds ranging from $1\times 10^{-5}$ to $1\times 10^{-1}$. **Bold** denotes better performance between our STL and MTL models.

Training Dataset (PLTD)		IJB-C (TAR@FAR)				
		$1\times 10^{-5}$	$1\times 10^{-4}$	$1\times 10^{-3}$	$1\times 10^{-2}$	$1\times 10^{-1}$
Ours (STL)	All Cases	**90.73**	94.05	96.28	97.89	98.95
	Soft Case	**89.03**	**92.5**	94.94	96.77	98.23
	Medium Case	76.41	84.19	89.31	93.51	96.47
	Hard Case	55.34	68.27	77.64	85.96	92.9
Ours (MTL)	All Cases	89.86	**94.17**	**96.6**	**97.95**	98.95
	Soft Case	87.18	92.42	**95.26**	**96.96**	**98.43**
	Medium Case	**77.21**	**86.03**	**91.27**	**94.57**	**96.99**
	Hard Case	**56.64**	**71.48**	**81.49**	**88.62**	**94.23**

**Table 5 sensors-24-03212-t005:** Mean absolute error (MAE) comparison on AFLW2000-3D [22] for head pose estimation. **Bold** denotes the best performance, and lower values reflect better results.

Method	Yaw	Pitch	Roll	MAE	FPS
FAN (12 points) [48]	6.36	12.3	8.71	9.12	-
HopeNet [75]	6.47	6.56	5.44	6.16	323.2
FSANet [44]	4.5	6.08	4.64	5.07	389.6
3DDFA-TPAMI [54]	4.33	5.98	4.30	4.87	-
3DDFA-V2 [53]	4.06	5.26	3.48	4.27	-
QuatNet [76]	3.97	5.62	3.92	4.15	-
TriNet [45]	4.2	5.77	4.04	3.97	311.3
RankPose [46]	2.99	4.75	3.25	3.66	291.7
SynergyNet [7]	3.42	**4.09**	**2.55**	3.35	128.1
Ours (Pretrained)	2.89	4.77	3.35	3.67	320.8
Ours (MTL)	**2.80**	4.27	2.9	**3.32**	221.2

**Table 6 sensors-24-03212-t006:** Mean absolute error (MAE) comparison on BIWI [42] for head pose estimation. **Bold** denotes best performance.

Method	Yaw	Pitch	Roll	MAE	FPS
3D-FAN [48]	8.53	7.48	7.63	7.89	-
HopeNet [75]	4.81	6.61	3.27	4.90	323.2
FSANet [44]	4.27	4.96	2.76	4.00	389.6
QuatNet [76]	4.01	5.49	2.94	4.15	-
TriNet [45]	4.11	**4.76**	3.05	3.97	311.3
RankPose [46]	3.59	4.77	2.76	3.71	291.7
Ours (pretrained)	4.34	5.18	2.61	4.04	320.8
Ours (MTL)	**3.23**	5.03	**2.36**	**3.54**	221.2

**Table 7 sensors-24-03212-t007:** Normalized mean square error (NMSE) percentage comparison on AFLW2000-3D [22] for 3D facial landmark detection across three yaw angle ranges, $0^{\circ }$ to $30^{\circ }$, $30^{\circ }$ to $60^{\circ }$, $60^{\circ }$ to $90^{\circ }$, and the overall mean error. **Bold** denotes the best performance, and lower values reflect better results.

Method	$0^{\circ }∼30^{\circ }$	$30^{\circ }∼60^{\circ }$	$60^{\circ }∼90^{\circ }$	Mean	FPS
3DSTN [77]	3.15	4.33	5.98	4.49	-
3D-FAN [48]	3.16	3.53	4.6	3.76	84.3
PRNet [78]	2.75	3.51	4.61	3.62	261.3
3DDFA-PAMI [54]	2.84	3.57	4.96	3.79	-
3DDFA-v2 [53]	2.63	3.42	4.48	3.51	-
SynergyNet [7]	2.65	**3.30**	**4.27**	**3.41**	128.1
Ours (pretrained)	3.08	3.74	4.53	3.78	322.5
Ours (MTL)	**2.62**	3.45	4.51	3.53	221.2

**Table 8 sensors-24-03212-t008:** Comparison of the performance of the baseline model in single-task learning (STL) and multitask learning (MTL) configurations across a variety of tasks, including face recognition (FR), head pose estimation (HPE), and facial landmark detection (FLD).

Method	FR	HPE	FLD	IJB-C	AFLW2000-3D (MSE)	AFLW2000-3D (MAE)	# Params	FPS
STL	✓	-	-	94.05	-	-	24.6M	321.7
STL	-	✓	-	-	3.67	-	23.5M	320.9
STL	-	-	✓	-	-	3.78	23.8M	322.5
MTL	✓	✓	-	94.16	3.54	-	56.8M	243.8
MTL	✓	-	✓	94.17	-	3.34	60.1M	237.9
MTL	-	✓	✓	-	3.52	3.33	47.3M	252.7
MTL	✓	✓	✓	94.17	3.53	3.32	73.3M	221.2

**Table 9 sensors-24-03212-t009:** Experimental results when changing the value of $\lambda_{P}$ in Equation (Equation 3) demonstrate the impact of the regularization intensity on the model performance across multiple benchmarks.

Method	IJB-C	AFLW2000-3D (NMSE)	AFLW2000-3d (MAE)	BIWI (MAE)
$L_{FLD}(\lambda_{P}=0$)	94.16	3.55	3.58	3.70
$L_{FLD}(\lambda_{P}=1$)	94.17	3.56	3.67	3.81
$L_{FLD}(\lambda_{P}=\alpha$)	94.17	3.53	3.32	3.54

**Table 10 sensors-24-03212-t010:** Experimental results showing the impact of multitask learning hyperparameters $\lambda_{1}$ and $\lambda_{2}$ on the test accuracy with various benchmarks. These hyperparameters are used in Equation (Equation 1) to balance the contributions of each task in the overall multitask learning framework. To check the change caused by $\lambda_{1}$ and $\lambda_{2}$, we obtained the loss convergence ratio based on the convergence value of the loss function of each task.

$\lambda_{1}$	$\lambda_{2}$	IJB-C ($1\times 10^{-4}$)	AFLW2000-3D (NMSE)	AFLW2000-3D (MAE)	Loss Convergence Ratio
0.1	0.1	94.22	5.12	4.14	0.98:0.00:0.02
0.1	1	94.21	4.77	3.42	0.86:0.00:0.14
0.1	10	94.22	4.56	3.35	0.37:0.00:0.63
0.1	100	94.23	4.98	3.35	0.08:0.00:0.92
1	0.1	94.20	4.61	3.98	0.96:0.01:0.03
1	1	94.19	4.21	3.41	0.85:0.01:0.14
1	10	94.12	4.37	3.39	0.37:0.01:0.62
1	100	94.18	4.27	3.38	0.08:0.01:0.91
10	0.1	94.88	3.99	3.94	0.94:0.05:0.02
10	1	94.19	3.65	3.54	0.82:0.04:0.14
10	10	94.17	3.61	3.33	0.35:0.02:0.63
10	100	94.17	3.60	3.33	0.06:0.00:0.93
100	0.1	94.18	3.74	3.99	0.67:0.32:0.01
100	1	94.19	3.55	3.59	0.61:0.29:0.10
100	10	**94.17**	**3.53**	**3.32**	0.31:0.15:0.54
100	100	94.17	3.54	3.32	0.04:0.03:0.94

## Data Availability

No new data were created or analyzed in this study. Data sharing is not applicable to this article.

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
