# Peer review of "Multitask Learning Strategy with Pseudo-Labeling: Face Recognition, Facial Landmark Detection, and Head Pose Estimation"

_sensors, 2024, doi:10.3390/s24103212_

Round 1

Reviewer 1 Report

Comments and Suggestions for Authors

Referee report

This paper is to study a pseudo-labeling technique using a face recognition dataset of various people and background noise. The results show that the proposed method achieves stable performance. Therefore, I think that the paper makes a contribution and has the potential to be published.  However, I summarize in the GENERAL COMMENTS as follows:

GENERAL COMMENTS

1. In Network Architecture section, the paper adopts ResNet50 and IResNet50, therefore the section provides the main features or processes of these frameworks.

2. In some formulas, such as formulas 6-7,12, mathematical variables should be italicized.

3. In formula 3, please provide the meaning of λP parameter.

4. Usually, the cumulative sum of weight coefficients is 1. Please provide your relevant theoretical basis and references. In formula (1), please provide a detailed explanation.

5. The results are not significant enough for journal publication yet. More comprehensive evaluations are needed for journal publication. In Table 3, since some methods provide results for MS1MV2, MS1MV3 datasets, the proposed algorithm should also be tested these datasets. Please provide results on MS1MV2 dataset.

Author Response

Response to Reviewer 1 Comments

1. Summary

Thank you for your insightful analysis and questions. Below, we have answered each of your questions and provided additional explanations and experimental results to clarify any gaps. Please note that due to time constraints, we were unable to conduct all the additional experiments, but we prioritized those that would add the most to the quality of the paper.

2. Point-by-point response to Comments and Suggestions for Authors

Comments 1: In Network Architecture section, the paper adopts ResNet50 and IResNet50, therefore the section provides the main features or processes of these frameworks.

Response 1: We agree that the description of the network structure was insufficient and thank you for pointing this out. The reason for using ResNet50 is that it is commonly used as a baseline for performance comparison in most face-related deep learning tasks. IResNet has some modifications from ResNet50, such as batch normalization, but these modifications are important for tasks that fall under the broader category of face analysis, such as face recognition, facial landmark detection, and head pose estimation. However, areas such as batch normalization and activation require detailed modifications and performance improvements, which is why we chose IResNet50 for multitask learning. Section 3.2.1 describes the network, but as you pointed out, it is not sufficiently descriptive, so we have revised it with the following additional text.

#page 11

To show that our proposed learning method is feasible, we chose ResNet[58 ], a network structure commonly used in face recognition-based networks, as a baseline and used the ResNet50 structure for our pretrained model to make the comparison as fair as possible. Furthermore, we used the modified IResNet50 [66 ], which is used in the recent face recognition-based methods such as ArcFace [4] and Partial FC [5], as the backbone of the multitask learning network. IResNet has no critical differences from ResNet, but we chose and modified the IResNet structure because it allows us to appropriately modify the details of the normalization and activation layers required by each task in face recognition, facial landmark detection, and head pose estimation. While tuning the network architecture for each task, we also considered maximizing shared parameters to minimize inference time loss. To do this, we referenced several face analysis tasks [7, 44 ,67 , 68] and finally split shared and independent parameters in a 3:1 ratio for each task in the baseline network. The modified IResNet50 consists of four layers, where the first three layers are shared layers, and the last layer is divided into sublayers customized for each task, as shown in Fig. 7.

Comments 2: In some formulas, such as formulas 6-7,12, mathematical variables should be italicized.

Response 2: We agree with the points and have made the necessary corrections. However, to distinguish vectors from variables in the various equations and algorithms presented in this manuscript, we have used a bold mathematical typeface for vectors.

For equation 12, we have modified the abbreviation for normalized mean square error (NMSE) as requested, and for the rest of the equations, we have not made any changes to distinguish between vectors and variables, but we will make changes in the future to make them less difficult to understand if it becomes necessary.

#page 18

[image] [shown in attached file]

Comments 3: In formula 3, please provide the meaning of λP parameter.

Response 3:

Thank you for your insightful comments. We recognize that an explanation for λP was missing in Formula 3 and have added further clarification. Formulas 6 through 9 continue to describe the process of calculating λP and its effects; however, we realized that an initial description of what the term represents was insufficient. We have made the following revisions accordingly.

#page 12

“We prepared two major loss functions for the loss function of facial landmark detection LFLD, as presented in Equation 3:

[equation] [shown in attached file]

where one is the Wing loss Lwing [46], and the other is the mean square error (MSE) loss multiplied by λP. λP is a regularization term that controls the convergence between the Wing loss and the MSE loss. The method for determining λP and its effects on the convergence of the losses are described in Equations 6 through 9. The wing loss Lwing is given by Equation 4:”

Comments 4: Usually, the cumulative sum of weight coefficients is 1. Please provide your relevant theoretical basis and references. In formula (1), please provide a detailed explanation.

Response 4: Thank you for your astute observation regarding the sum of the weight coefficients. In our paper, we specified that in Formula (1), the coefficient for the Face Recognition Loss value was set to 1, while the coefficients for the Facial Landmark Detection and Head Pose Estimation were determined experimentally. An example of this approach in the literature is the use of weight values in loss functions to adjust the emphasis on learning specific parts or referencing styles in studies involving datasets or tasks across different domains, such as in CycleGAN[*1]. Whether or not the sum of weight coefficients equals one, the influence can be modulated throughout the training process by adjusting the learning rate. This makes it meaningful to set a baseline task and conduct experiments to adjust the other factors accordingly. In this paper, with a dataset comprising over 8 million images, where more than 95% are used for Face Recognition, we set the coefficient value for this loss function to 1 and applied weight coefficients to the relatively volatile tasks of facial landmark detection and head pose estimation. This was to ensure that the learning rate has a uniform impact on the training of each task, setting lambda 1 and lambda 2 as the final convergence criteria.

However, we agree that the explanation for the optimal ranges of lambda 1 and 2 obtained through experimentation was not satisfactory, and additional experiments have been conducted in response. In Table 10, we added a new row for lambda coefficients, which documents the ratios of L_FR, L_FLD, and L_HPE values at final convergence, and further explanations have been added on page 19~20. We believe that the newly added row and descriptions will demonstrate that a higher learning proportion of each loss function can lead to significant performance achievements in their respective tasks. The modifications can be reviewed in the text below.

#page 11

where LFR, LFLD, and LHPE denote the loss functions of face recognition, facial landmark detection, and head pose estimation, respectively. We set the base criterion at 1 for LFR and applied the learning weights λ1 and λ2 to LFLD and LHPE, respectively. These weights were adjusted based on the sum of the values represented by the final output of each task. We fine-tuned each task’s contribution to the overall multitask framework by choosing λ1 to be 100 and λ2 to be 10 to provide an appropriate learning ratio based on the final convergence of each task’s loss function. We created our multitask loss function in Equation 1 by computing the total loss value Ltotal by summing all weighted losses from each task.

#page 20

 In multitask learning, adjusting the learning weights for the loss function designed for each task is essential for the optimization process. In this paper, we take the loss function for face recognition as the baseline in the unified loss function designed in Equation 1 and set the hyperparameter coefficients of λ1 and λ2 to the other tasks, facial landmark detection and head pose estimation, respectively. Table 10 shows the results of experiments conducted to find the optimal hyperparameters. In detail, the loss convergence ratio is the ratio of the values of LFR, λ1 ∗ LFLD, and λ2 ∗ LFLD in equation 1 based on the loss value of each task when learning converges to the maximum. The results show that the convergence ratio should be distributed to a certain extent to converge to more optimal multitask learning. Learning by increasing the hyperparameter coefficients for the remaining two tasks is more effective in our proposed multitask learning framework than learning by expanding the hyperparameter coefficients for face recognition, which constitutes most of the pseudo-labeled training dataset.

[table] [shown in attached file]

[*1] Zhu, Jun-Yan, et al. "Unpaired image-to-image translation using cycle-consistent adversarial networks." Proceedings of the IEEE international conference on computer vision. 2017.

Comments 5: The results are not significant enough for journal publication yet. More comprehensive evaluations are needed for journal publication. In Table 3, since some methods provide results for MS1MV2, MS1MV3 datasets, the proposed algorithm should also be tested these datasets. Please provide results on MS1MV2 dataset.

Response 5: I will respond to your last comment. First, I agree that the selection process for the dataset described in Section 3.1 may not have been sufficiently convincing.

Regarding the use of the MS-Celeb-1M dataset versions 1, 2, and 3, I would like to explain why we could not use all versions.

Firstly, the MS-Celeb-1M dataset version 1 had privacy issues related to the rights of individuals' portraits, and there were overlapping concerns with version 2 during that period, which made us hesitant to use these versions. Version 3 addressed these issues with the help of RetinaFace[*1], and was officially provided at the IEEE/CVF International Conference on Computer Vision Workshops, 2019 (ICCV2019); hence, we opted to use this version.

Secondly, when initially setting up the experiments, we tested both versions 2 and 3; however, to conduct all the experiments proposed in the paper, the pseudo dataset configuration would take 10 days, and each experiment conducted with version 3 took 3 to 6 days, requiring about an additional month of experimental time.

Therefore, we kindly ask for your understanding regarding our choice to use the MS1MV3 dataset for the face recognition task. A review of recent papers related to face recognition and their benchmarks within those papers shows that there is a trend to use only the MS1MV3 dataset from MS-Celeb-1M.

[*1] Deng, Jiankang, et al. "Lightweight face recognition challenge." Proceedings of the IEEE/CVF International Conference on Computer Vision Workshops. 2019.

Reviewer 2 Report

Comments and Suggestions for Authors

The author proposes a pseudo-labeling technique using a face recognition dataset of various people and background noise. The proposed method achieves near-SOTA for multi-task learning using this technique.

Overall, the paper is pretty good, with extensive experiments and ablation studies. I have some suggestions as follows.

(1) How can the cosine similarity distance threshold for the pseudo labeling generation process be selected? Is that sensitive to the final performance?

(2) Pseudo-labeling is a very promising direction, and there are several works related to pseudo-labeling for image classification and segmentation problems. Please add the following reference in your introduction section, which will help readers understand the potential of this field.

[1] Lee, D.-H.Pseudo-label: The simple and efficient semi-supervised learning method for deep neural networks. In Proceedings of the Workshop on Challenges in Representation Learning, International Conference on Machine Learning, Atlanta, GA, USA, 16–21 June 2013; Volume 3.

[2] Yu, X., Ouyang, B., Principe, J. C., Farrington, S., Reed, J., & Li, Y. (2019, October). Weakly supervised learning of point-level annotation for coral image segmentation. In OCEANS 2019 MTS/IEEE SEATTLE (pp. 1-7). IEEE.

[3] Caron, Mathilde, et al. "Deep clustering for unsupervised learning of visual features." Proceedings of the European conference on computer vision (ECCV). 2018.

Author Response

Response to Reviewer 2 Comments

1. Summary

Thank you for your insightful analysis and questions. Below, we have answered your questions and provided additional explanations. The revised manuscript also contains more experimental results to clarify it.

2. Point-by-point response to Comments and Suggestions for Authors

Comments 1: How can the cosine similarity distance threshold for the pseudo labeling generation process be selected? Is that sensitive to the final performance?

Response 1: We acknowledge and appreciate your point regarding the lack of explanation for the threshold values used in the pseudo-labeling process. In this manuscript, there are three primary thresholds used.

The first is the Euclidean distance threshold, , used in Algorithm 1, which employs the average performance value from the test to detect outliers among multiple pseudo landmark samples. This is discussed in the manuscript.

We realized that the other two thresholds lacked specific explanations, so we've written more about them in Section 3.1.3. These two thresholds are derived from the pre-training of the face recognition task on the VGGFace2 and MS1MV3 datasets, respectively. They can typically be measured using the LFW verification test, a common validation test in face recognition. Therefore, the thresholds,  and,  were selected to maximize performance in the LFW verification test for each dataset.

There are generally two usable distance metrics: measuring either the Euclidean distance or the angular distance between feature vectors. The choice of distance metric depends on whether the Euclidean or angular distance was set as the loss function. Using a different distance metric than the one set during training can significantly reduce performance. Hence, we can only use cosine similarity in the pseudo-labeling framework, as we employed a large margin cosine loss function that utilizes angular distance during pre-training.

We could have used Euclidean distance, but the trend of recent face recognition tasks such as ArcFace and PartialFC mentioned in the manuscript shows that angular distance is better for feature clustering than Euclidean distance. We chose to use the angular distance metric in our experiments to keep the performance of all tasks as high as possible due to the multi-task learning process.

#page 9

We have described how to label pseudo-landmarks and pseudo-pose labels. Next, we describe a framework for distributing face IDs to integrate the dataset in the following algorithm 2. First, most FR datasets are composed of data on celebrities and other famous individuals. Consequently, some duplicate IDs exist between the FR datasets. To filter out duplicates between the VGGFace2 [28] and MS1MV3 [29] datasets, we calculated the similarity threshold values tMS and tVGG while pretraining the face recognition task with each dataset. During the pretraining process, we used the LFW [65] verification test, commonly employed to evaluate the performance of face recognition tasks, to measure the optimal similarity threshold that maximizes verification performance. This threshold was then used as the similarity threshold for detecting duplicates in algorithm 2. Both angular and Euclidean distances can be used to measure similarity; however, the same distance metric must be employed in the loss function used during pretraining. Since we used the large margin cosine loss [38] function in our proposed method, we employed angular distance for cosine similarity in detecting data duplicates. For multitask learning, we filtered out duplicate IDs between VGGFace2 [28] and MS1MV3 [29 ] and added new IDs to 300W-LP [22] to pseudo-labeled training dataset.

Comments 2: Pseudo-labeling is a very promising direction, and there are several works related to pseudo-labeling for image classification and segmentation problems. Please add the following reference in your introduction section, which will help readers understand the potential of this field.

[1] Lee, D.-H.Pseudo-label: The simple and efficient semi-supervised learning method for deep neural networks. In Proceedings of the Workshop on Challenges in Representation Learning, International Conference on Machine Learning, Atlanta, GA, USA, 16–21 June 2013; Volume 3.

[2] Yu, X., Ouyang, B., Principe, J. C., Farrington, S., Reed, J., & Li, Y. (2019, October). Weakly supervised learning of point-level annotation for coral image segmentation. In OCEANS 2019 MTS/IEEE SEATTLE (pp. 1-7). IEEE.

[3] Caron, Mathilde, et al. "Deep clustering for unsupervised learning of visual features." Proceedings of the European conference on computer vision (ECCV). 2018.

Response 2: Thank you for your understanding of the potential of pseudo-labeling and for recommending related work. We've added each reference to this manuscript in the related work and introduction. You can find them below.

.

#page 2

“However, these approaches assume label availability for multiple tasks within a dataset. Consequently, they inherently suffer from challenges as a fundamental solution to the lack of face IDs and overall volume in datasets. Alternative approaches exist, such as improving data quantity through research in pseudo-labeling methods [ 22–26 ]. Inspired by multitask learning methods such as all-in-one CNN [ 27 ] and SwinFace [ 21 ] as well as pseudo-labeling methods [22 ,23 ] such as those proposed by Pan et al., we hypothesized that using datasets such as those for face recognition tasks that have numerous images with a variety of face IDs would be feasible for a task with a scarce number of face IDs.”

Reviewer 3 Report

Comments and Suggestions for Authors

The work deals with the well-known task of face recognition, as well as head pose recognition and key point extraction of faces. The main difference is the reduction of the data to more realistic conditions. The authors manage to obtain interesting results for the scientific community through the use of pseudolabels. However, there are a number of comments to the paper.

1) In the review part, we should mention the works devoted to optimizing not only the quality but also the processing speed (10.3103/S87566990210101012X, 10.1007/s11633-023-1423-y), as well as the application of simple neural networks and the study of their quality under different parameters (10.1134/S1054661821030032)

2) Line 189: Paragraph (3.1.1) cannot start with a figure. Preliminary text needs to be added.

3) Figure 3: Text extends beyond the borders (green box) on the bottom left. Needs to be fixed.

4) It would be interesting if the authors added training plots for the loss function they introduced.

5) Performance (FPS) data for tables 3, 4 and 5(6) should be added.

Author Response

Response to Reviewer 3 Comments

1. Summary

Thank you very much for taking the time to review this manuscript. Please find the detailed responses below and the corresponding revisions/corrections highlighted/in track changes in the re-submitted files. [This is only a recommended summary. Please feel free to adjust it. We do suggest maintaining a neutral tone and thanking the reviewers for their contribution although the comments may be negative or off-target. If you disagree with the reviewer's comments please include any concerns you may have in the letter to the Academic Editor.]

2. Point-by-point response to Comments and Suggestions for Authors

Comments 1: In the review part, we should mention the works devoted to optimizing not only the quality but also the processing speed (10.3103/S87566990210101012X, 10.1007/s11633-023-1423-y), as well as the application of simple neural networks and the study of their quality under different parameters (10.1134/S1054661821030032)

Response 1: We agree that not only quality but also processing speed should be given important weight in facial analysis task, and thank you for introducing several related works. We could not add the 10.3103/S87566990210101012X paper because we could not confirm it, but for the other two papers, we used them as references in the network architecture section (Section 3.2.1).

#page 11

“To show that our proposed learning method is feasible, we chose ResNet[58], a network structure commonly used in face recognition-based networks, as a baseline and used the ResNet50 structure for our pretrained model to make the comparison as fair as possible. Furthermore, we used the modified IResNet50 [66], which is used in the recent face recognition-based methods such as ArcFace [4] and Partial FC [5], as the backbone of the multitask learning network. IResNet has no critical differences from ResNet, but we chose and modified the IResNet structure because it allows us to appropriately modify the details of the normalization and activation layers required by each task in face recognition, facial landmark detection, and head pose estimation. While tuning the network architecture for each task, we also considered maximizing shared parameters to minimize inference time loss. To do this, we referenced several face analysis tasks [7, 44, 67, 68] and finally split shared and independent parameters in a 3:1 ratio for each task in the baseline network. The modified IResNet50 consists of four layers, where the first three layers are shared layers, and the last layer is divided into sublayers customized for each task, as shown in Fig. 7.”

Comments 2: Line 189: Paragraph (3.1.1) cannot start with a figure. Preliminary text needs to be added.

Response 2: Thank you for your keen observation and response. We moved the sentence to start first at the beginning of the section.

#page 5

3.1.1. Dataset selection strategy Table 1 lists the commonly used datasets for each task. We focus on the ratios of the datasets, number of images per ID, and total amount of data. We selected three datasets as the pseudo-labeled training dataset: two FR datasets, VGGFace2 [ 28 ] and MS1MV3 [ 29],
and 300W-LP [22], which is the labeled dataset for HPE and FLD.

Comments 3: Figure 3: Text extends beyond the borders (green box) on the bottom left. Needs to be fixed.

Response 3: Thank you for your comments on this aspect. The text at the bottom of Figure 3 was intended to be outside of the figure, but as you mentioned, it looks more like it's sticking out, so we fixed it.

#page 7

[image] [shown in attached file]

Comments 4: It would be interesting if the authors added training plots for the loss function they introduced.

Response 4: Thank you for your suggestion about adding a training plot. We thought it was significant to show the training convergence curve of each loss function, but due to the experiments with coefficient variation for each task analyzed in Table 10, we felt that there were too many labels to plot, so we added the loss convergence ratio of each task during the final training process to Table 10.

#page 11

where LFR, LFLD, and LHPE denote the loss functions of face recognition, facial landmark detection, and head pose estimation, respectively. We set the base criterion at 1 for LFR and applied the learning weights λ1 and λ2 to LFLD and LHPE, respectively. These weights were adjusted based on the sum of the values represented by the final output of each task. We fine-tuned each task’s contribution to the overall multitask framework by choosing λ1 to be 100 and λ2 to be 10 to provide an appropriate learning ratio based on the final convergence of each task’s loss function. We created our multitask loss function in Equation 1 by computing the total loss value Ltotal by summing all weighted losses from each task.

#page 20

In multitask learning, adjusting the learning weights for the loss function designed for each task is essential for the optimization process. In this paper, we take the loss function for face recognition as the baseline in the unified loss function designed in Equation 1 and set the hyperparameter coefficients of λ1 and λ2 to the other tasks, facial landmark detection and head pose estimation, respectively. Table 10 shows the results of experiments conducted to find the optimal hyperparameters. In detail, the loss convergence ratio is the ratio of LFR, λ1 ∗ LFLD, and λ2 ∗ LFLD in equation 1 based on the loss value of each task when learning converges to the maximum. The results show that the convergence ratio should be distributed to a certain extent to converge to more optimal multitask learning. Learning by increasing the hyperparameter coefficients for the remaining two tasks is more effective in our proposed multitask learning framework than learning by expanding the hyperparameter coefficients for face recognition, which constitutes most of the pseudo-labeled training dataset.

[table] [shown in attached file]

[*1] Zhu, Jun-Yan, et al. "Unpaired image-to-image translation using cycle-consistent adversarial networks." Proceedings of the IEEE international conference on computer vision. 2017.

Comments 5: Performance (FPS) data for tables 3, 4 and 5(6) should be added.

Response 5: Thank you for your thoughtful comment adding performance for FPS. We wanted to calculate and add FPS for all of the comparative papers, but we were not able to add all of them, and for those references that had executable code, we measured the FPS performance and reflected it in each table.

However, we excluded table 4 because we thought it would be better to cover it in table 8, and we added its FPS to table 8.

We also added a brief analysis of it, and we think that this comparison of FPS performance helps the reader to understand a little more the research direction that we want to present in this manuscript.

#page 16

[table] [shown in attached file]

These results indicate that filtering and learning from them is feasible even if the pseudo-labeled dataset incorporates some outliers. It also exhibits the potential to effortlessly integrate various facial analysis tasks beyond the proposed methods, highlighting the versatility and adaptability of our approach despite anomalies and discrepancies within the data. In addition, we compared the inference time with similar papers under identical conditions and conducted a frame-per-second (fps) comparison for the ResNet50 and IResNet50 models used in the comparison papers. Unlike the recent trend of using arrays of multiple GPUs for training face recognition methods, our fps comparison was conducted using just a single GPU. Despite performing multitask computations, our proposed method performed well in terms of fps.

#page 17

[table] [shown in attached file]

#page 18

 [table] [shown in attached file]

#page 19

 [table] [shown in attached file]

#page 19

 [table] [shown in attached file]

Round 2

Reviewer 3 Report

Comments and Suggestions for Authors

The remarks have been corrected, the work is ready for publication.